# Perforin and Granzyme B Expressed by Murine Myeloid-Derived Suppressor Cells: A Study on Their Role in Outgrowth of Cancer Cells

**DOI:** 10.3390/cancers11060808

**Published:** 2019-06-11

**Authors:** Inès Dufait, Julian Pardo, David Escors, Yannick De Vlaeminck, Heng Jiang, Marleen Keyaerts, Mark De Ridder, Karine Breckpot

**Affiliations:** 1Laboratory of Translational Radiation Oncology Physics and Supportive Care, Department of Radiotherapy, UZ Brussel, Vrije Universteit Brussel, Laarbeeklaan 101, 1090 Brussels, Belgium; ines.dufait@vub.be (I.D.); jiangheng1981@gmail.com (H.J.); mark.deridder@uzbrussel.be (M.D.R.); 2Laboratory of Molecular and Cellular Therapy, Department of Biomedical Sciences, Vrije Universiteit Brussel, Brussels, Laarbeeklaan 103, 1090 Brussels, Belgium; yannick.de.vlaeminck@vub.be; 3Fundación Instituto de Investigación Sanitaria de Aragón/ Universidad de Zaragoza, Centro de Investigación Biomédica de Aragón, Calle de San Juan Bosco, 13, 50009 Zaragoza, Spain; pardojim@unizar.es; 4Fundación Aragon I+D (ARAID), Universidad de Zaragoza, Aragón, Calle de San Juan Bosco, 13, 50009 Zaragoza, Spain; 5Immunomodulation group, Navarrabiomed-Biomedical Research Centre, IdISNA. C/irunlarrea 3 Complejo Hospitalario de Navarra, 31008 Pamplona, Navarra, Spain; davidescors@gmail.com; 6Rayne Institute, Division of Infection and Immunity, University College London. 5 University St, London WC1E 6JF, UK; 7In Vivo Cellular and Molecular Imaging Laboratory, Vrije Universiteit Brussel, Laarbeeklaan 103, 1090 Brussels, Belgium; marleen.keyaerts@vub.be; 8Nuclear Medicine Department, UZ Brussel, Laarbeeklaan 101, 1090 Brussels, Belgium

**Keywords:** MDSC, CD8+ T-cell, perforin, granzyme B, cancer

## Abstract

A wide-range of myeloid-derived suppressor cell (MDSC)-mediated immune suppressive functions has previously been described. Nevertheless, potential novel mechanisms by which MDSCs aid tumor progression are, in all likelihood, still unrecognized. Next to its well-known expression in natural killer cells and cytotoxic T lymphocytes (CTLs), granzyme B (GzmB) expression has been found in different cell types. In an MDSC culture model, we demonstrated perforin and GzmB expression. Furthermore, similar observations were made in MDSCs isolated from tumor-bearing mice. Even in MDSCs from humans, GzmB expression was demonstrated. Of note, B16F10 melanoma cells co-cultured with perforin/GzmB knock out mice (KO) MDSCs displayed a remarkable decrease in invasive potential. B16F10 melanoma cells co-injected with KO MDSCs, displayed a significant slower growth curve compared to tumor cells co-injected with wild type (WT) MDSCs. In vivo absence of perforin/GzmB in MDSCs resulted in a higher number of CD8^+^ T-cells. Despite this change in favor of CD8^+^ T-cell infiltration, we observed low interferon-γ (IFN-γ) and high programmed death-ligand 1 (PD-L1) expression, suggesting that other immunosuppressive mechanisms render these CD8^+^ T-cells dysfunctional. Taken together, our results suggest that GzmB expression in MDSCs is another means to promote tumor growth and warrants further investigation to unravel the exact underlying mechanism.

## 1. Introduction

Myeloid-derived suppressor cells (MDSCs) are a heterogeneous population of immature myeloid cells that are, in large, defined by their ability to inhibit T-cell responses. Phenotypically, MDSCs are characterized by the expression of CD11b and Gr-1. Subdivision in monocytic (M) and polymorphonuclear (PMN) MDSCs is based on two structurally similar molecules, Ly6C and Ly6G, respectively [1].

Within the tumor microenvironment (TME), MDSCs pose a major hurdle, since these cells exert immunosuppressive functions and are implicated in providing cues that promote tumor cell survival and metastasis [2]. To exert their immunosuppressive function, MDSCs exploit a number of mechanisms, among which is expression of arginase-1 (Arg-1) and inducible nitric oxide synthase (iNOS), production of interleukin-10 (IL-10) and transforming growth factor-β (TGF-β), and induction of regulatory T-cells (Tregs) [3]. To promote tumor cell survival and metastasis, M-MDSCs and PMN-MDSCs have distinct roles that act in concert. While M-MDSCs act on the tumor side, inducing epithelial to mesenchymal transition (EMT) of cancer cells, PMN-MDSCs act at the side of cancer cell colonization. Therefore, both MDSC-subsets display distinct molecular profiles, with M-MDSCs characterized by the expression of IL-1a, IL-6, TGF-β1, and iNOS and PMN-MDSCs characterized by the expression of S100 proteins, matrix metalloproteinase (MMPs), and TGF-β3 [4].

Granzymes (Gzms) are serine proteases that could be implicated in the immunosuppressive and metastasis-promoting function of MDSCs. Together with perforin, GzmB is a key component of the lytic machinery of cytotoxic T lymphocytes (CTLs) and natural killer (NK) cells. Several studies reported the expression of GzmB in other cell types in both human and mouse, including B cells, mast cells, basophils, macrophages, blood PMN cells, dendritic cells (DCs), keratincoytes, pneumocytes, and cells of the reproductive system and cartilage [5,6,7,8,9,10,11,12,13,14]. Immature DCs expand using the granulocyte-macrophage colony stimulating factor (GM-CSF) express perforin and Gzms and kill CD8^+^ T-cells in a cell contact-dependent way by a mechanism dependent on perforin [15]. As GM-CSF is a driving force in the expansion and accumulation of MDSCs, it is conceivable that these cells also express perforin and Gzms to inhibit CD8^+^ T-cell responses. Moreover, there is ample evidence of perforin-independent functions of GzmB that can be linked to tumor metastasis [14]. These functions include proteolytic activity to facilitate the pro-inflammatory activity of IL-1a, which is implicated in the ability of M-MDSCs to promote EMT of cancer cells, and to cleave extracellular matrix (ECM) proteins, a function usually linked to MMPs and implicated in the ability of PMN-MDSCs to enable cancer cell colonization in distant organs [4,16,17,18,19,20].

Therefore, we studied the expression of perforin and GzmB by MDSCs and its functional implications.

## 2. Results

### 2.1. Perforin and GzmB are Expressed by MDSCs

We generated in vitro MDSCs that closely resemble those found within the melanoma TME by culturing bone marrow cells in a conditioned medium (CM) of B16F10 cells that were modified to secrete high levels of GM-CSF [21,22]. We performed flow cytometry to detect perforin and GzmB in both M- and PMN-MDSCs, defined as CD11b^+^Ly6C^high^Ly6G^low^ and CD11b^+^Ly6C^low^Ly6G^high^ cells, respectively. High expression of perforin and GzmB was observed in both MDSC-subsets. As a control, MDSCs were generated from bone marrow of perforin/GzmB knock out mice (KO) mice, lacking perforin and GzmB, showing only background staining of perforin and GzmB (Appendix A).

To ensure that the expression of perforin and GzmB was not model-dependent, we next generated MDSCs that closely resemble those found within colorectal carcinoma. Bone marrow cells were cultured in a CM of CT26 cells that were modified to secrete high levels of GM-CSF [22]. Perforin and GzmB expression was detected in both M- and PMN-MDSCs (Figure 1b).

To ensure that the expression of perforin and GzmB detected in in vitro MDSCs is not an artifact of the in vitro culture and is representative for different tumor models, we studied MDSCs isolated from the tumor and spleen of mice bearing B16F10 melanoma, CT26 colorectal carcinoma, E.G7-OVA T-cell lymphoma, and 4T1 mammary carcinoma. The flow cytometry showed that tumor- and spleen-MDSCs expressed perforin and GzmB (Figure 2a,b).

To assess the value of these findings, we next analyzed the expression of perforin and GzmB in M-MDSCs (CD11b^+^CD33^+^CD14^+^HLA-DR^low^) and PMN-MDSCs (CD11b^+^CD33^+^CD15^+^) of colon cancer patients and healthy donors. We could not observe the expression of perforin compared to the isotype control with the used antibody; however, we observed that both MDSC-subsets expressed high levels of GzmB in both colon cancer patients and healthy donors (Figure 2c).

### 2.2. Perforin and GzmB Expressing MDSCs Promote Tumor Progression 

Since GzmB can exert both perforin-dependent and -independent functions, we further studied the functional relevance of perforin and GzmB expression by murine MDSCs [4,16,17,18,19,20]. First, we thoroughly compared the MDSCs generated from the bone marrow of wild type (WT) and KO mice. We did not observe differences in the phenotype (Figure 3a) or expression of Arg-1 and iNOS (Figure 3b) between WT and KO MDSCs. Moreover, we studied the expression of MMP9 as its expression by MDSCs has been linked to their tumor-promoting potential [23]. A gelatin zymography assay revealed a similar MMP expression by WT and KO MDSCs (Figure 3c). These results suggest that any effects observed in vivo could be due to the effect of perforin and GzmB on the MDSCs’ ability to facilitate tumor growth.

Next, we performed an in vivo experiment in which B16F10-Fluc cells were co-injected with MDSCs generated from the bone marrow of WT or KO mice using CM collected from B16F10-GM-CSF cells. In vivo bioluminescence imaging, which can detect early tumor onset as the B16-F10 cells express Fluc, was used to follow up the primary tumor growth rate and possible formation of metastasis (Figure 4a). The tumor onset (day five) was not affected by the co-injection of B16F10-Fluc cells with WT versus KO MDSCs (Figure 4b). However, we observed that tumor cells inoculated together with WT MDSCs had a growth advantage over tumor cells inoculated without MDSCs or with KO MDSCs with the majority of tumors reaching 500 mm^3^ on day 14, at which time the tumors generated by co-injection with KO MDSCs were on average 122 ± 98 mm^3^, as measured by caliper (Figure 4c). Primary tumors were resected when they reached 502 ± 103 mm^3^, and occurrence of metastasis was followed by in vivo bioluminescence imaging, expecting an in vivo bioluminescent signal at regions where B16F10-Fluc cells grow. However, metastases were not observed in any of the conditions, despite a follow up time of 30 days.

### 2.3. GzmB Expressed by MDSCs Facilitates Migration of Tumor Cells 

To study the mechanisms underlying the enhanced tumor outgrowth, we first studied whether GzmB enhances the migratory capacity of tumor cells by degradation of the ECM. Therefore, WT MDSCs, with or without the addition of a GzmB inhibitor (Z-AAD-CMK) or KO MDSCs were co-cultured with B16F10 cells and subjected to a scratch wound migration assay, in which matrigel was used as a substitute of the ECM. We observed that only the condition wherein B16F10 cells were cultured in the presence of WT MDSCs showed a statistical significant higher migratory capacity than B16F10 cells cultured in the absence of MDSCs, or in the presence of KO MDSCs or WT MDSCs and a GzmB inhibitor, as evidenced by the gap width measured 48 h later, which was 505 ± 117 µm, 629 ± 23 µm, 583 ± 34 µm, and 543 ± 25 µm, respectively (Figure 5a,b).

### 2.4. MDSCs Expressing Perforin and Granzyme B Affect CD8^+^ T-cell Numbers in the TME

Since MDSCs express both GzmB and perforin, and as these effector molecules expressed by immature DCs and Tregs were described to exert cytolytic effects on NK and T-cells [15,24], we next studied whether MDSCs use a similar mechanism to suppress T-cells.

We first performed an in vitro co-culture assay to study the survival, proliferation, and function of T-cells when cultured with KO or WT MDSCs either in or not in the presence of the GzmB inhibitor, Z-AAD-CMK. However, we did not observe significant differences for any of the evaluated parameters between T-cells cultured in the presence of WT MDSCs, with or without the GzmB inhibitor or KO MDSCs (Figure 6a–c).

Since we observed a difference in tumor growth in vivo between tumor cells inoculated together with WT and KO MDSCs, we further studied tumor-infiltrating T-cells using this experimental set up. Flow cytometry on tumors that were resected when reaching 502 ± 103 mm^3^ showed that B16F10-Fluc tumors in mice co-injected with WT MDSCs had a significant lower percentage of CD8^+^ T-cells (3.3 ± 1.3%) when compared to B16F10-Fluc tumors that were co-injected with KO MDSCs (10.6 ± 4.1%), while the percentage of CD3^+^ and CD4^+^ T-cells were unchanged (Figure 6d–f). We further evaluated the expression of interferon-γ (IFN-γ) by real-time polymerase chain reaction (RT-PCR) and compared the results to those of tumors initially grown in the absence of MDSCs. We observed an increase in IFN-γ expression (4.6 ± 3.0) in B16F10-Fluc tumors of mice co-injected with WT MDSCs, while a decrease was observed in B16F10-Fluc tumors co-injected with KO MDSCs (0.5 ± 0.5) (Figure 6f). As this result is seemingly contradictory, we hypothesized that KO MDSCs co-injected with B16F10-Fluc tumor cells initially do not hamper CD8^+^ T-cells as much as WT MDSCs do, enabling these CD8^+^ T-cells to keep the tumor growth under control during the first days of tumor growth (Figure 4b), and likely pressuring B16F10-Fluc tumor cells to adopt mechanisms to paralyze these CD8^+^ T-cells. Therefore, we evaluated the expression of Programmed Death-Ligand 1 (PD-L1) by RT-PCR in the TME, as this is a well-known inhibitory immune checkpoint axis implicated in adaptive resistance. We observed an increased PD-L1 expression in the KO group (5.6 ± 3.5), while PD-L1 expression in the WT group (1.0 ± 1.0) was unchanged when compared to the PD-L1 expression in B16F10-Fluc tumors that were initially grown in the absence of MDSCs.

## 3. Discussion

Perforin and Gzm activity are implicated in several physiological functions, ranging from cardiovascular disease over wound healing to involvement in inflammation and cancer [25,26,27,28,29,30]. This led to the understanding that Gzm activity is complex and background- and cell type-dependent. Additionally, its role in onco-immunology becomes generally more accepted, although both protective and detrimental effects have been described [14]. In this study, we addressed the role of GzmB and perforin in the context of MDSCs, a previous unexplored field concerning Gzm biology in onco-immunology.

We showed that in vitro MDSCs and MDSCs isolated from in vivo grown tumors express GzmB and perforin, thereby confirming that the in vitro MDSC platform can be used to discover new targets regarding MDSC biology [21,22]. We moreover showed that MDSCs isolated from peripheral blood of colon cancer patients express GzmB. However, we could not detect positive staining for perforin with the used antibody in these human MDSCs. This might be explained by the observation in preclinical models that the MDSCs isolated from the spleen (periphery) and tumor show significant differences in expression of functional molecules [22,31]. Also in this study, we observed in the 4T1 breast carcinoma, CT26 colorectal carcinoma, and E.G7-OVA T-cell lymphoma model that expression of perforin in MDSCs isolated from the spleen was significantly lower, and in the E.G7-OVA T-cell lymphoma was even undetectable when compared to the MDSCs isolated from the respective tumors. However, this was not the case for the B16 melanoma model. A possible hypothesis for this discrepancy is the intrinsic differences between different tumor models. However, it is undeniable that tumor-infiltrating myeloid cells behave differently than circulating cells [32]. Therefore, it remains to be addressed whether MDSCs found in peripheral blood and colon tumors in patients are positive for perforin. Nevertheless, our data confirm previous observations of GzmB expression in MDSCs in patients suffering from myelodysplastic syndrome [13]. Together with the description of perforin and GzmA and GzmB expression in immature DCs [15], this suggests that GzmB and perforin expression in myeloid cells is not only related to tumor development, but can be considered a general feature of myeloid cells.

We hypothesized that GzmB expressed by MDSCs could be implicated in the immunosuppressive- and migration-promoting function of MDSCs by direct killing of T-cells and degradation of the ECM, respectively. To study the tumor promoting properties of perforin and GzmB expressing MDSCs, we first performed an in vivo assay, evaluating the growth and metastasis of B16F10 cells, an aggressive melanoma tumor model. Standard in vivo metastasis models with B16F10 cells are obtained by injecting tumor cells intravenously. However, degradation of the ECM to facilitate tumor dissemination is not required, therefore this model is not suited for our purpose, which aspires to take into account the effects on immune cells and tumor cell spreading by degradation of the ECM. We consequently considered using KO mice. However, tumors do not develop well in these mice [24,33]. Moreover, the lack of perforin and GzmB in this model is not confined to MDSCs, thus, any conclusions drawn could be confounded by other perforin and GzmB expressing immune cell populations, such as Tregs, CTLs, and NK cells [5,24]. Therefore, we settled on a previously described approach, in which tumor cells are co-injected with MDSCs [34,35]. This set up has the drawback that it only takes the effects of GzmB and perforin expressed by MDSCs into account at early stages of tumor development, which likely explains why we were unable to show any effect on metastasis, while in vitro we showed that WT MDSCs facilitate migration of tumor cells towards a scratch wound in contrast to KO MDSCs or WT MDSCs exposed to a GzmB inhibitor. As both WT and KO MDSCs expressed similar levels of MMPs, and as the addition of a GzmB inhibitor sufficed to reverse the migratory capacity, this effect is most likely attributed to the expression of GzmB and not perforin by WT MDSCs and gives a first clue to what is happening in vivo. This was expected, as GzmB has been implicated in the cleavage of no less than 10 proteins found within the ECM [36].

The in vivo tumor growth experiment further showed that the co-injection of WT or KO MDSCs did not affect the tumor onset. However, we observed that the outgrowth of tumors when tumor cells are inoculated together with WT MDSCs is at a faster pace when compared to the outgrowth of tumors when solely tumor cells are inoculated [34,35]. When injecting tumor cells with KO MDSCs, this growth advantage was lost. These data show that WT MDSCs affect the early TME in a different way than the KO MDSCs. Because CD8^+^ T-cells are well known for their tumor controlling capacities, and several studies linked the expression of perforin and/or GzmA and GzmB in DCs, human plasmacytoid DCs, and murine conventional DCs to the inhibition and even killing of T-cells in vitro [15,37,38], we first performed an in vitro co-culture between T-cells and WT or KO MDSCs to study the differences in T-cell suppression. In contrast to the studies described with DCs, we were unable to show any effects on T-cell survival, proliferation, or production of IFN-γ. The latter might be explained by the experimental set up, as it was shown that perforin and Gzm-mediated killing of mouse CD8^+^ T-cells depended on the recognition of mismatched major histocompatibility complex (MHC) molecules [15]. We performed our experiments in an autologous set up, because this was most relevant with respect to the in vivo experiment. Another explanation could be that, in this experimental set up, in which T-cells were activated via anti-CD3/anti-CD28 antibody coated beads and therefore were likely to express high levels of perforin and GzmB, the effects due to a lack of these molecules in MDSCs is masked. Based on this in vitro assay, we would conclude that there are no direct negative effects of perforin and GzmB expressed by MDSCs on T-cells, which is in contrast to studies with other perforin- and GzmB-expressing immune cells, in particular Tregs, which show that these effector molecules contribute to the killing of NK cells and CTLs [24]. Therefore, we examined the presence of CD8^+^ T-cells in the TME of B16F10-Fluc tumors that developed after injection of B16F10-Fluc tumor cells together with WT or KO MDSCs. We observed that there were significant differences in the CD8^+^ T-cell infiltrate of these tumors with a higher number of CD8^+^ T-cells when B16F10-Fluc tumor cells were co-injected with KO MDSCs. It has been described that GzmB expressed by CTLs facilitates the transmigration of CTLs’ through blood vessels [39]. Although this same study did not link GzmB expression by CTLs to infiltration in established tumors, it prompted us to speculate that GzmB expressed by MDSCs at the onset of tumor development facilitates the transmigration of CD8^+^ T-cells by making the ECM leaky. The change in CD8^+^ T-cells but not CD4^+^ T-cell numbers, moreover, provokes us to speculate that in vivo CD8^+^ T-cells are deleted by perforin- and GzmB-expressing MDSCs, which is in line with the observation that CD8^+^ T-cells can be killed in a perforin- and Gzm-dependent fashion by immature DCs, while this is not the case for CD4^+^ T-cells [15].

The enhanced number of CD8^+^ T-cells observed in B16F10-Fluc tumors of mice inoculated with B16F10-Fluc cells and KO MDSCs is suggestive for the role of CD8^+^ T-cells in the observed delay in outgrowth of tumors. In contradiction, herewith is the observation that IFN-γ is significantly lower expressed when compared to B16F10-Fluc tumors of mice inoculated with B16F10-Fluc cells and WT MDSCs. We hypothesized that, because of the higher percentage of CD8^+^ T-cells in the B16F10-Fluc tumors of mice inoculated with B16F10-Fluc cells and KO MDSCs, tumor cells would be under pressure and would likely adapt resistance mechanisms to circumvent CD8^+^ T-cell-mediated killing, such as the expression of PD-L1 [40]. In addition, PD-L1 expressed by cancer cells directly inhibits IFN-γ-mediated apoptosis and accelerates their tumor growth in vivo [41]. This could explain the presence of high numbers of relative inactive CD8^+^ T-cells. When evaluating the expression of PD-L1, we observed that PD-L1 expression was significantly higher in B16F10-Fluc tumors of mice inoculated with B16F10-Fluc cells and KO MDSCs when compared to B16F10-Fluc tumors of mice inoculated with B16F10-Fluc cells and WT MDSCs, corroborating our hypothesis.

## 4. Materials and Methods 

### 4.1. Cell Lines and Mice

B16F0, B16F10, CT26, 4T1, and E.G7-OVA cells were obtained from American Type Culture Collection (ATCC) and were cultured, following ATCC’s recommendations. The B16F10-GM-CSF and CT26-GM-CSF cell lines were generated and cultured as described [21,22]. The vector encoding a thermostable and red shifted Firefly luciferase was described [42]. This vector was used to stably modify B16F10 cells (B16F10-Fluc) as described for R1M cells [43]. Perforin and GzmB knock out mice (KO) were obtained from Julian Pardo (Universidad de Zaragoza, Spain). Female C57BL/6 and Balb/c mice (7–8 weeks) were purchased from Charles River Laboratories (L’Arbresle Cedex, France).

Animals were handled according to the institutional guidelines, with the approval of the Ethical Committee for use of laboratory animals of the Vrije Universiteit Brussel (Belgium) (approval code 17-214-2).

### 4.2. In Vitro Differentiation of Myeloid-Derived Suppressor Cells

The conditioned medium (CM) was harvested from B16F10-GM-CSF and CT26-GM-CSF cells and used to differentiate bone marrow cells into MDSCs as described [21]. Briefly, 10 × 10^6^ bone marrow cells were cultured for 6 days in 75% CM and 25% Iscove’s Modified Dulbecco’s medium (Sigma–Aldrich, Diegem, Belgium) supplemented with 10% fetal clone I (GE Health Care Life Sciences, Utah, USA), 100 U/mL penicillin, 100 μg/mL streptomycin, and 2 mM L-glutamine (Sigma–Aldrich, Diegem, Belgium).

### 4.3. Isolation of Human Myeloid Cells

Peripheral blood was collected from 5 colorectal cancer patients and 5 healthy donors. Buffy coat was separated using Histopaque solution (Sigma–Aldrich, Diegem, Belgium), according to manufacturer’s instructions. This study was approved by the ethical committee of the UZ Brussels (approval code: B.U.N. 143201628651). Written informed consent was obtained from all the subjects.

### 4.4. In Vivo Tumor Growth

To obtain tumors for characterization of in vivo MDSCs, 3 × 10^5^ tumor cells were transferred subcutaneously at the right flank of C57BL/6 (B16-F10 and E.G7-OVA) or Balb/c (CT26 and 4T1) mice. Tumor growth was followed on a daily basis and the tumor size measured using an automated caliper. Tumors of 502 ± 103 mm^3^ were isolated and reduced to single cell suspensions using the GentleMACs protocol (Miltenyi Biotec, Bergisch-Gladbach, Germany). MDSCs present in these cell suspensions were characterized by flow cytometry. To study tumor cell growth and metastasis, 3 × 10^5^ B16F10-Fluc cells were inoculated subcutaneously at the right flank of C57BL/6 mice with or without 1 × 10^5^ in vitro MDSCs of wild type (WT) or KO mice. Tumor onset and growth was followed using in vivo bioluminescence imaging. The volume of the subcutaneous tumors was measured using an automated caliper. Subcutaneous tumors were removed when reaching on average 502 ± 103 mm^3^, after which formation of metastasis was followed using in vivo bioluminescence imaging. The resected tumors were cut in two. It was part-snap-frozen for RNA extraction and part-digested into a single cell suspension as described above.

### 4.5. In Vivo Bioluminescence Imaging

In vivo bioluminescence imaging was performed at different time points after the tumor inoculation as previously described [43]. Briefly, mice were anaesthetised with a mixture of oxygen/isoflurane, 5% isoflurane for induction, and 2.5% isoflurane for maintenance, using an inhalation anesthesia system (VetTech solutions, Congleton, UK). D-luciferin was injected intraperitioneally at 150 mg/kg mouse body weight (Promega, Madison, WI, USA). Immediately after D-luciferin administration, mice were imaged using the photo imager (Biospace, Urbandale, Iowa, USA). The photon emission was measured dynamically using the large field-of-view setting and registered using the photon counting technology (Biospace) during 10 min. A photographic image was obtained at the end of each acquisition, and bioluminescent pseudocolor images were superimposed on these gray-scale photographic images. The most intense Fluc signal is shown as red, while the weakest signal is shown as blue. For image analysis, an elliptical region of interest (ROI) was drawn over the tumor location. The surface area of the ROI was kept constant and the quantification was performed at when a plateau was reached using a constant time interval, being 12 to 15 min after injection of D-luciferin.

### 4.6. Quantitative RT-PCR

Snap-frozen tumors were disrupted using 6 mm stainless steel beads in the TissueLyser (Qiagen, Hilden, Germany). RNA was extracted using the 6100 Nucleic acid prepstation (Applied Biosystems, California, USA), according to the manufacturer’s instructions. copyDNA was synthesized using random primers in a thermal cycler (Applied Biosystems). Primers specific for mouse hypoxanthine phosphoribosyltransferase 1 (HPRT1), programmed death-ligand 1 (PD-L1), and interferon-γ (IFN-γ) were used and relative gene expression was determined using the iQ SYBR Green supermix-CFX Connect Real-Time System (BioRad Laboratories, Temse, Belgium). The comparative threshold cycle method was used to calculate gene expression normalized to HPRT1 as a reference gene.

### 4.7. In Vitro T-cell Assay

An in vitro T-cell assay was performed as described to evaluate the MDSC’s activity [22]. Briefly, 1 × 10^5^ carboxy fluorescein diacetate succinimidyl ester (Invitrogen, California, USA) labeled T-cells were activated with a 1/800 dilution of anti-CD3/CD28 antibody coated beads (Invitrogen, California, USA) and cultured with or without 5 × 10^4^ WT or KO MDSCs. When indicated, the GzmB inhibitor, Z-AAD-CMK (Enzo Life Sciences, Antwerpen, Belgium), was added (10 μM). T-cell proliferation and viability were determined in flow cytometry.

### 4.8. Flow Cytometry

The staining of cell surface markers and annexin V/7-AAD was performed as described [44]. For staining of intracellular perforin and GzmB, cells were treated with inside FIX (eBioscience, Vienna, Austria) and incubated for 20 min at room temperature. Cells were further incubated with PERM (eBioscience) and the staining antibody for 20 min at room temperature. Subsequently, cells were washed with a FACS buffer. For staining of intracellular IFN-γ, cells were treated with Brefeldin A (BD Biosciences) for 4 h before intracellular staining. Subsequently, cells were treated with inside FIX (eBioscience) and incubated for 20 min at room temperature. Cells were further incubated with PERM (eBioscience) and the staining antibody for 20 min at room temperature. Subsequently, cells were washed with a FACS buffer. The antibodies used throughout this study are listed in Table 1. Cells stained with isotype-matched control antibodies served as a control. Cells were acquired using the LSR Fortessa (Becton Dickinson, Erembodegem, Belgium), and analyzed using FlowJo 7.6 (Treestar Inc, Oregon, USA).

### 4.9. In Vitro Invasion Assay

The IncuCyte Zoom Scratch Wound assay (Essen Bioscience) was used to examine the invasive potential of B16F10 cells in the presence of MDSCs. First, 96-well plates (ImageLock) were coated overnight at 37 °C with 100 µg/mL matrigel (Corning, New York, USA). Next, wells were seeded with 9×10^5^ B16F10 cells. After 6 h, scratches were introduced using the IncuCyte WoundMaker^TM^ (Essen Bioscience, Welwyn Garden City, UK). B16F10 cells and the induced scratches were covered with 50 μL matrigel (3 mg/mL in culture medium). When indicated, an equal amount of WT or KO MDSCs was added in matrigel. This matrix was overlaid with additional 100 μL of culture medium. Wound confluency was monitored every 2 h for a total of 48 h with the IncuCyte Live Cell Imaging System (Essen Bioscience). The IncuCyte scratch wound analysis software was used to measure the wound width.

### 4.10. Gelatin Zymography

Gelatin zymography was used to measure the activity of MMPs as described [45]. Briefly, samples (serum free) were subjected to electrophoresis in a 7.5% gel containing 0.1% gelatin, followed by incubation overnight in an incubation buffer at 37 °C to activate the enzymatic activity. The gel was stained with 0.1% Coomassie blue and afterwards destained. White bands were visualized by Odyssey and analyzed using Image Studio Lite (Li-core Biosciences, Lincoln, NE, USA).

### 4.11. Statistical Analyses

One- or two-way ANOVA, followed by Bonferroni’s multiple comparison or t-tests, were performed using Graphpad Prism6.1 (Graphpad Software, Suite, San Diego, CA, USA). The number of experimental repetitions and statistical significances are indicated in the figure legends.

## 5. Conclusions

We demonstrated that MDSCs express perforin and GzmB, and that these effector molecules contribute to the MDSCs ability to promote tumor progression in multiple ways, namely enhanced invasive capability of tumor cells, enhanced tumor growth, and the ability to influence CD8^+^ T-cell responses. This observation adds to our understanding of the regulatory role of MDSCs in cancer, providing yet another piece of evidence supporting the notion that targeting of MDSCs will be indispensable to win the fight against cancer.

## Figures and Tables

**Figure 1 cancers-11-00808-f001:**
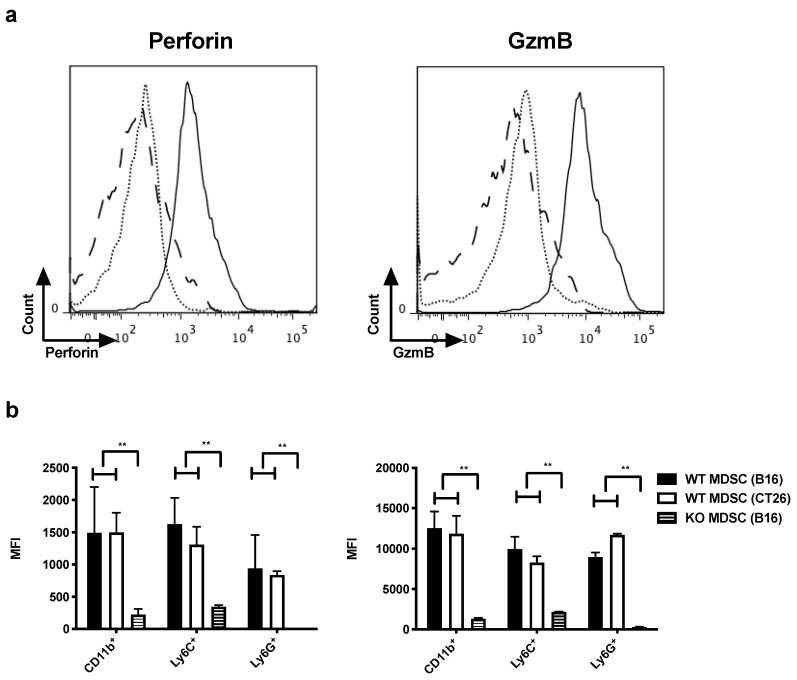
In vitro MDSCs express perforin and GzmB. (**a**,**b**) MDSCs were generated starting from bone marrow of WT or perforin/GzmB knock out mice (KO) mice using the CM of tumor cells. (**a**) Representative histograms showing perforin (left) and GzmB (right) expression by in vitro WT (full line) or KO (dotted line) MDSCs (gated on CD11b^+^ cells) compared to the isotype control (long dashed line), generated using the CM of B16F10-GM-CSF cells. (**b**) Summarizing graphs showing the mean fluorescence intensity (MFI) of perforin (left) and GzmB (right) of in vitro MDSCs (CD11b^+^) and of M-MDSCs (CD11b^+^Ly6C^+^) and PMN-MDSCs (CD11b^+^Ly6G^+^), generated using the CM of B16F10-GM-CSF cells (WT MDSC (B16) and KO MDSC (B16)) and the CM of CT26-GM-CSF cells (WT MDSC (CT26)). The mean +/- standard error of the mean (SEM) of at least three experiments is shown. A student’s t-test was used to calculate statistical significance. The number of asterisks in the figures indicates the level of statistical significance as follows: **, *p* < 0.01.

**Figure 2 cancers-11-00808-f002:**
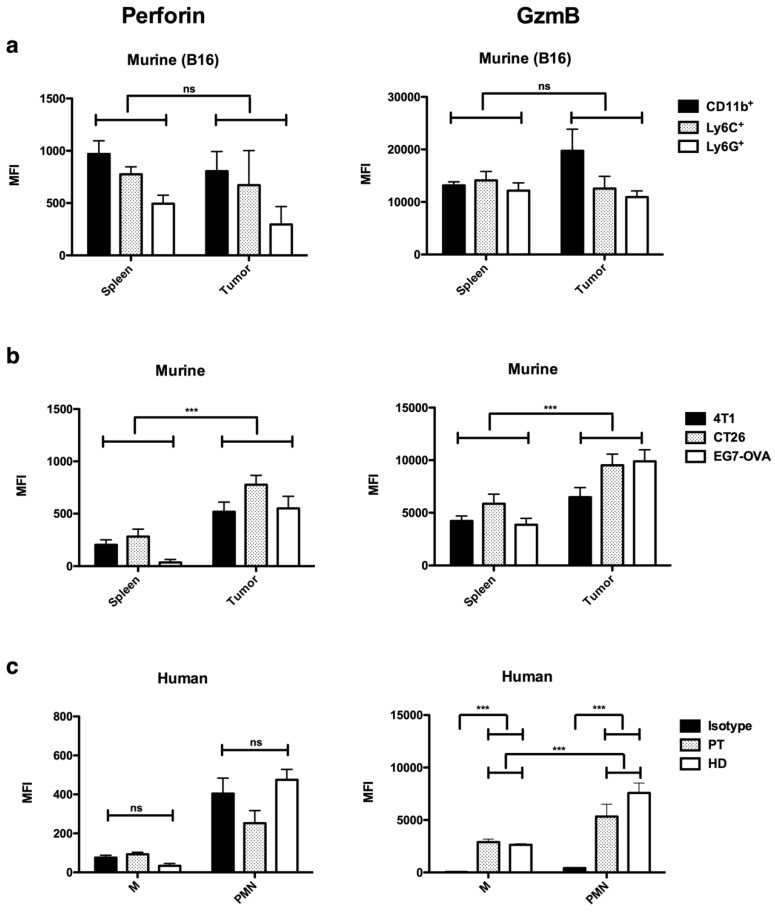
In vivo MDSCs express perforin and GzmB, while human circulating myeloid cells only express GzmB. (**a**) Summarizing graphs showing the MFI of perforin (left) and GzmB (right) in MDSCs (CD11b^+^) and both MDSC subsets (Ly6C^+^ and Ly6G^+^) isolated from the tumor and spleen of B16F10-bearing mice. (**b**) Summarizing graphs showing the MFI of perforin (left) and GzmB (right) in MDSCs (CD11b^+^) isolated from the spleen and tumor in mice bearing different tumor cell lines. The mean +/- SEM of at least 3 experiments is shown in all graphs. A two-way ANOVA was used to calculate statistical significance. (**c**) Summarizing graphs showing the MFI of perforin (left) and GzmB (right) in peripheral blood M- (CD11b^+^CD33^+^CD14^+^HLA-DR^low^) and PMN-MDSC (CD11b^+^CD33^+^CD15^+^) from colorectal cancer patients (PT) and healthy donors (HD) compared to isotype control (Isotype). The mean +/-SEM of at least five data points is shown in all graphs. A student’s t-test was used to calculate the statistical significance. The number of asterisks in the figures indicates the level of statistical significance as follows: ns, >0.05 and ***, *p* < 0.001.

**Figure 3 cancers-11-00808-f003:**
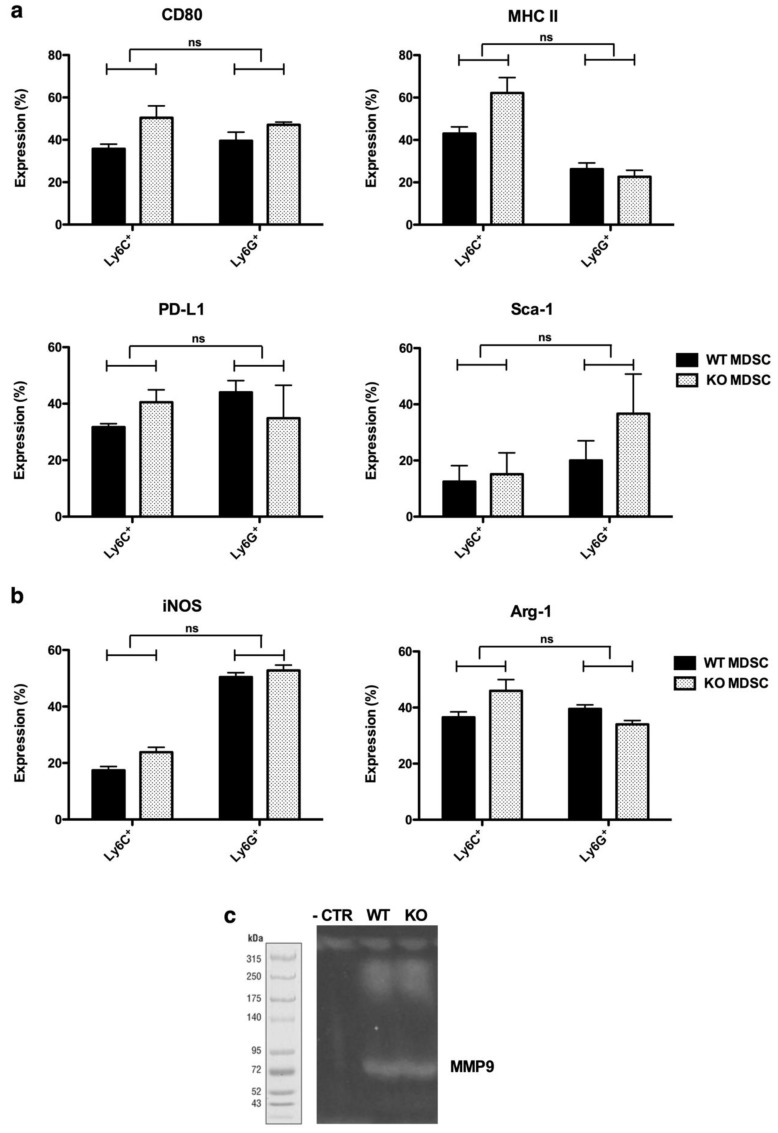
In vitro WT and KO MDSCs show similar properties. (**a**) Summarizing graphs showing the expression of different surface markers (CD80, MHC II, programmed death-ligand 1 (PD-L1), and Sca-1) on WT and KO MDSCs, gated by CD11b^+^. Expression showed in M-MDSCs (Ly6C^+^) and PMN-MDSCs (Ly6G^+^) separately. (**b**) Summarizing graphs showing the expression of functional markers (inducible nitric oxide synthase (iNOS) on the left and arginase-1 (Arg-1) on the right) on WT and KO MDSCs. The mean +/-SEM of at least three experiments is shown in all graphs. A student’s t-test was used to calculate the statistical significance. The number of asterisks in the figures indicates the level of statistical significance as follows: ns, >0.05 and *, *p* > 0.05. (**c**) Representative image of gelatin zymography assay showing MMP9 activity in WT and KO MDSCs.

**Figure 4 cancers-11-00808-f004:**
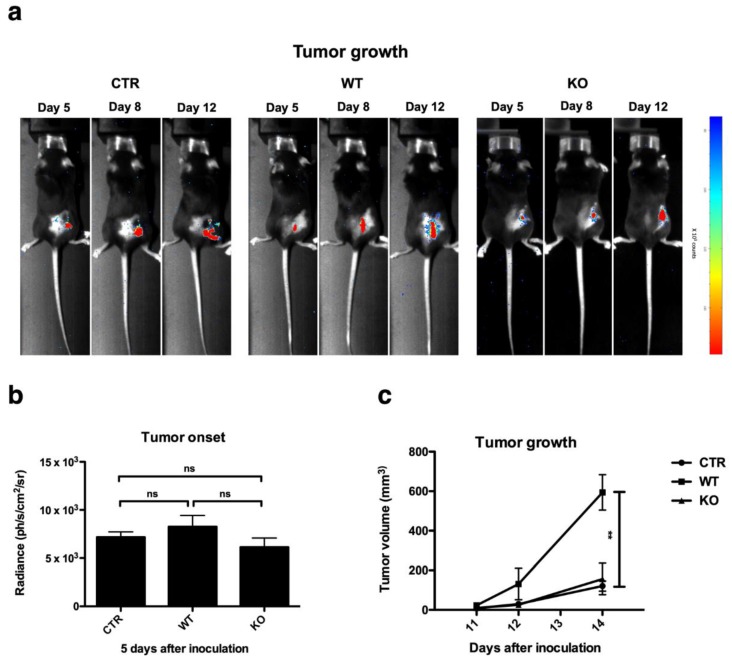
Perforin and GzmB expressing MDSCs promote tumor progression. (**a**) Representative images showing tumor growth of mice inoculated with B16F10-Fluc cells alone, as a control (CTR) or B16F10-Fluc cells co-injected with WT or KO MDSCs, as measured by in vivo bioluminescence imaging. (**b**) Summarizing graph showing onset of tumor growth five days after injection of B16F10-Fluc cells (CTR), or co-injection of B16F10-Fluc cells and WT or KO MDSCs, as measured by in vivo bioluminescence imaging. (**c**) Representative tumor growth curve of mice inoculated with B16F10-Fluc cells alone (CTR) or co-injected with WT or KO MDSCs, as measured by caliper. The graph was shown until day 14, when all tumors in the WT group were resected for further analysis. The mean +/-SEM of at least six data points is shown in all graphs. All experiments were repeated at least once. A one-way ANOVA was used to calculate statistical significance. Number of asterisks in the figures indicates the level of statistical significance as follows: ns, > 0.05 and **, *p* < 0.01.

**Figure 5 cancers-11-00808-f005:**
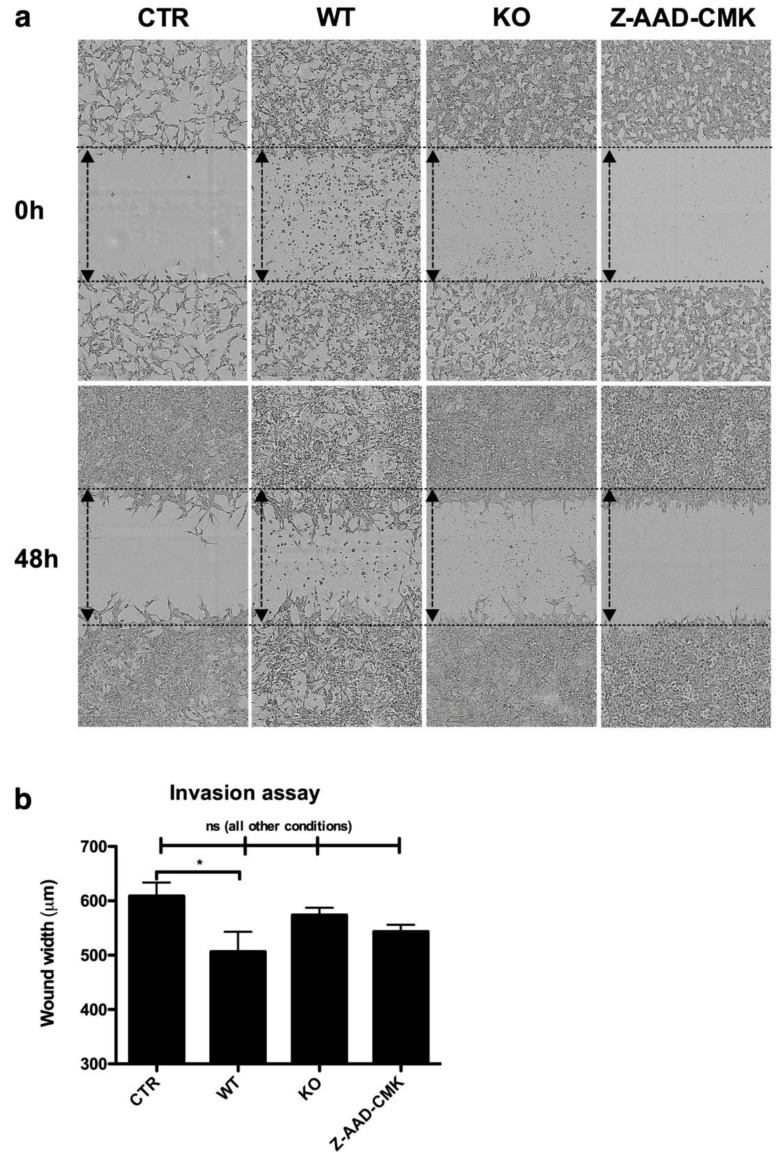
GzmB expressed by MDSCs facilitate migration of tumor cells. (**a**) Representative images of the in vitro invasion assay after 48 h incubation. (**b**) Summarizing graph of the in vitro invasion assay showing the wound width after 48 h of incubation. The following conditions were shown: B16F10 cells alone (CTR), B16F10 cells supplemented with WT MDSCs (WT), B16F10 cells supplemented with KO MDSCs (KO), and B16F10 cells supplemented with WT MDSCs and the GzmB inhibitor Z-AAD-CMK (10 μM) (Z-AAD-CMK), respectively. The mean +/-SEM of at least fivw data points is shown. A student’s t-test was used to calculate the statistical significance. The number of asterisks in the figures indicates the level of statistical significance as follows: ns, > 0.05 and *, *p* < 0.05.

**Figure 6 cancers-11-00808-f006:**
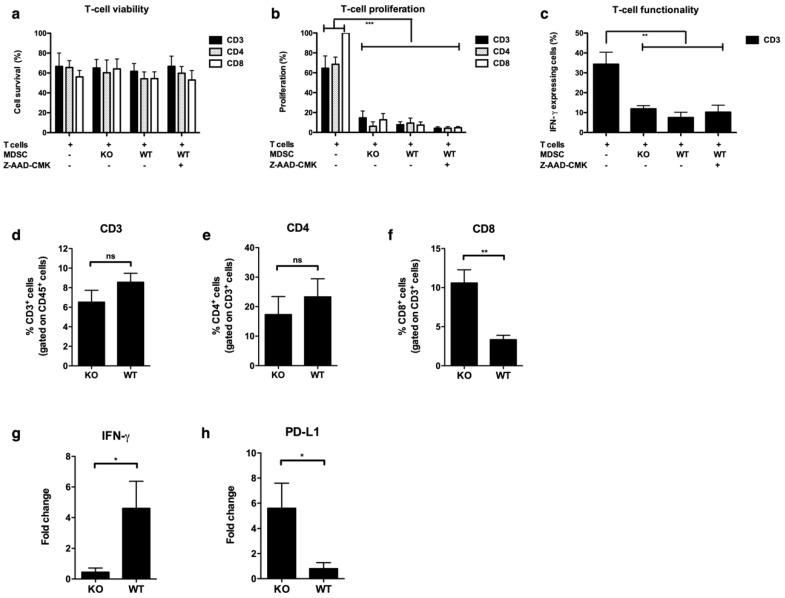
Perforin and GzmB expressing how MDSCs affect CD8^+^ T-cell numbers in the tumor microenvironment (TME). (**a**–**c**) Carboxyfluorescein succinimidyl ester (CFSE)-labeled T-cells were activated with anti-CD3/CD28 antibody coated beads and cultured without MDSCs, with KO MDSCs or with WT MDSCs in the presence or absence of the GzmB inhibitor, Z-AAD-CMK. A MDSC:T-cell ratio of 1:2 was used. (**a**) Summarizing graph showing the viability of T-cells measured by flow cytometry as 7-AAD^−^/Annexin V^+^ cells. (**b**) Summarizing graph showing the inhibition of T-cell proliferation measured by flow cytometry as reduced CFSE dilution. (**c**) Summarizing graph showing interferon-γ (IFN-γ) secretion by T-cells measured by flow cytometry. The mean +/-SEM of at least three experiments is shown in all graphs. For (**a**) and (**b**), a two-way ANOVA was used to calculate the statistical significance, while for (**c**), a one-way ANOVA was used. The number of asterisks in the figures indicates the level of statistical significance as follows: **, *p* < 0.01 and ***, *p* < 0.001. (**d**–**f**) Summarizing graphs showing the % of (**d**) CD3, (**e**) CD4, or (**f**) CD8 T-cells, as measured by the flow cytometry in single cell suspensions of tumors (502 ± 103 mm^3^) that were grown in vivo starting from B16F10-Fluc cells co-injected with WT or KO MDSCs. (**g**–**h**) Summarizing graph showing the fold change, as measured by quantitative PCR of (**g**) IFN-γ and (**h**) Programmed Death- Ligand 1 (PD-L1) expression tumors (502 ± 103 mm^3^) that were grown in vivo, starting from B16F10-Fluc cells co-injected with WT or KO MDSCs. The fold change was normalized to B16F10-Fluc tumors that were grown after injection of tumor cells alone. The graphs show the mean ±SEM of at least three data points. Experiments were performed twice. A student’s t-test was used to calculate the statistics. Number of asterisks in the figures indicates the level of statistical significance as follows: ns, >0.05, *, *p* < 0.05 and **, *p* < 0.01.

**Table 1 cancers-11-00808-t001:** Specifications of the used antibodies.

Species	Marker	Fluorochrome	Company
Anti-human	CD11b	BV421	BD Biosciences
	CD14	APC-eF780	eBioscience
	CD15	PerCP-eF710	
	CD33	PeCy7	
	HLA-DR	APC	
	Granzyme B	FITC	Thermofisher
	Perforin	PE	Abcam
Anti-Mouse	CD3	FITC	BD Biosciences
	CD11b	FITC	
	CD3	PerCP-Cy5.5	eBioscience
	CD4	eF450	
	CD8	APC-eF780	
	CD11b	eF450	
	CD45	eF450	
	Granzyme B	PerCP-eF710	
	Perforin	PE	
	Sca-1	AF700	
	Annexin V	PE	
	7-AAD		
	CD80	BV421	Biolegend
	IFN-γ	PeCy7	
	Ly6G	AF647	
	Ly6C	PeCy7	
	MHC II	PE	
	PD-L1	PE	
Anti-human/mouse	Arg-1	PE	R&D
	iNOS	PercP-Cy5.5	Santa Cruz

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
