# Peer review of "Perforin and Granzyme B Expressed by Murine Myeloid-Derived Suppressor Cells: A Study on Their Role in Outgrowth of Cancer Cells"

_cancers, 2019, doi:10.3390/cancers11060808_

Round 1

Reviewer 1 Report

In this study authors tried to prove that perforin and Granzyme B from MDSCs are key factors that promote tumor progression. They inoculated tumor cells with MDSCs from Perforin/Granzyme KO mice and evaluated tumor growth and metastasis. Major concerns are the protocol for MDSCs generation and the method for MDSCs characterization. Although they differentiate MDSCs from bone marrow cells of WT and KO mice follow their established protocol at Oncotarget 2014 using condition medium (CM) of tumor cell culture. They should demonstrate the percentage of CD11b, Ly6C or Ly6G population in different kinds of CM. Thus they can dissect and make sure the expression of perforin and Granzyme B are majority from MDSCs instead of other CD11b/Ly6C or CD11b/Ly6G double negative population. In addition, there are still numerous concerns should be addressed:

1. Figure 1A, the background or isotype control staining need to show using overlay at histogram. The population and cell count of MDSCs should demonstrate. And the alive% quantification should be described as they describe Annexin V/7-AAD staining at Method 4.8. Need to specify the purpose of using B16F10-GM-CSF, but not B16F10.

2. The author validated the perforin and granzyme B expression using flow cytometry. There is no information related to intracellular staining or surface staining of these two molecules. Also, they need to show dot blot or histogram raw data and the dead cell rule out strategy.

3. Figure 3, they differentiate MDSCs from KO mice bone marrow cells. Are these KO mice having both perforin and granzyme B double knockout? Seems Fig. 2 clear describe that there is perforin independent role of human MDSCs, the difference of perforin and granzyme might need to perform using single knockout mice. 

4. Figure. 4, authors checked tumor and growth curve at day 5 and around 2 weeks. Due to the in vivo study they use was co-injected of tumor with MDSCs, whether perforin and granzyme B effects directly through MDSCs or indirectly via tumor secreting factors? MDSCs conditional knockout in vivo model and transwell assay should be considered to address direct or contact dependent effect of MDSCs.  

Author Response

Reviewer 1:

Review Report Form:

Yes

Can be improved

Must be improved

Not applicable

Does the introduction provide sufficient background and include all relevant references?

( )

(x)

( )

( )

Is the research design appropriate?

( )

(x)

( )

( )

Are the methods adequately described?

( )

( )

(x)

( )

Are the results clearly presented?

( )

(x)

( )

( )

Are the conclusions supported by the results?

( )

( )

(x)

( )

Reply: We thank the reviewer for reading and commenting on the manuscript, submitted to “Cancers”. We have done everything possible before the revision deadline of April 29, 2019 to comply with the reviewer’s comments, and provide a more in depth description of these changes in the point-by-point reply to the comments and suggestions of the reviewer.

Comments and Suggestions for Authors:

In this study, the authors tried to prove that perforin and Granzyme B from MDSCs are key factors that promote tumor progression. They inoculated tumor cells with MDSCs from Perforin/Granzyme KO mice and evaluated tumor growth and metastasis. Major concerns are the protocol for MDSC generation and the method for MDSC characterization. Although they differentiate MDSCs from bone marrow cells of WT and KO mice follow their established protocol at Oncotarget 2014 using condition medium (CM) of tumor cell culture. They should demonstrate the percentage of CD11b, Ly6C or Ly6G population in different kinds of CM. Thus they can dissect and make sure the expression of perforin and Granzyme B are majority from MDSCs instead of other CD11b/Ly6C or CD11b/Ly6G double negative population.

Reply: We understand the reviewer’s skepticism on the use of in vitro generated MDSCs and whether or not these cultures are heterogenous. In this regard, we wish to explain that the culture system was optimized by Therese Liechtenstein, a member of the research team of David Escors, and that different steps in the optimzation protocol, the resulting cell yield, heterogeneity of the culture, suppressive activity of the cells has been extensively tested (using CM of several cell lines). The reference to that work is Liechtenstein et al, 2014, Oncotarget, Vol. 5, Number 17, Page: 7843. In short, it was shown that the optimized condition, yields high levels of CD11b positive cells, which show the following phenotype (Figure below). Compared to immature dendritic cells, the MDSCs show little CD11c expression, and reduced expression of many of the antigen-presenting or co-stimulatory molecules typically found on dendritic cells (even macrophages). 

It was also confirmed that the phenotype of these cells was similar to that of tumor-derived MDSCs (Figure below). This figure moreover shows that MDSC isolated from the spleen, as used in many studies, are less representative than the in vitro MDSC system used in ours studies to examine the biology of MDSCs.

We also evaluated the suppressive activity of in vitro generated MDSCs and showed that both the granulocytic (Ly6G high) and monocytic subset (Ly6C high) are highly suppressive even when using a MDSC:T cell ratio of 1:8.

Subsequently, in vitro generated MDSCs (using the optimized protocol of Therese Liechtenstein) were used to (1) discover new anti-neoplastic targets, (2) reveal that P450R expression in melanoma-specific MDSCs sensitized them to Paclitaxel treatment while it protected MDSCs against other chemotherapy drugs such as Irinotecan (Liechtenstein et al, 2014, Oncotarget, Vol. 5, Number 17, Page: 7843), (3) predict which vaccines, containing factors that inhibit the suppressive effect of MDSC on T cells, are effective in vivo (Liechtenstein et al, 2014, Oncoimmunology, Vol. 3, Number 7, Page: e945378), (3) to study the effect of a novel fusokine on MDSCs and link this to its in vivo efficacy to delay tumor growth (Van der Jeught et al, 2014, Oncotarget, Vol. 5, Number 20, Page: 10100), (4) to study MDSCs in the context of colorectal cancer (Dufait et al, 2016, Vol. 6, Number 14, Page: 12369). These studies have extended the use of CM generated from melanoma cells to CM of other cancer types, among others the colorectal cancer cell line CT26.

Because of these findings and the ethical issues related to use of tumor-derived MDSCs for which large groups of tumor-bearing animals are required to obtain sufficient MDSC, we are in favor of using this in vitro MDSC differentiation system. We also feel that because we have published this system multiple times (incl. phenotyping in every occasion) it is redundant to show the phenotype, as the figures we have on the MDSCs used in this study show similar expression of CD111b, Ly6C, Ly6G, etc, therefore, are “copies” of the published figures.

There are still numerous concerns that should be addressed:

1.  Figure 1A, the background or isotype control staining needs to be shown using overlay at histogram. The population and cell count of MDSCs should be demonstrated. And the alive% quantification should be described as they describe Annexin V/7-AAD staining at Method 4.8. Need to specify the purpose of using B16F10-GM-CSF, but not B16F10.

Reply: As mentioned above, we would like to refrain from showing the MDSC cell count, phenotype and viability, as this information is similar to the already published data (publications in which Karine Breckpot is co-author or senior author).

As requested, we have added the histogram showing the isotype control in Figure 1a.

As requested, we have clarified in section 4.8, why B16F10-GM-CSF cells were used.

2.  The author validated the perforin and granzyme B expression using flow cytometry. There is no information related to intracellular staining or surface staining of these two molecules. Also, they need to show dot blot or histogram raw data and the dead cell rule out strategy.

Reply: As requested by the reviewer, we provide additional information on the staining procedure in the methods section. We furthermore provided a figure showing the gating strategy (see Figure below). This figure was included in the manuscript as well.

3.  Figure 3, they differentiate MDSCs from KO mice bone marrow cells. Are these KO mice having both perforin and granzyme B double knockout? Seems Fig. 2 clear describe that there is perforin independent role of human MDSCs, the difference of perforin and granzyme might need to perform using single knockout mice.  

Reply: The mice used in this study are indeed perforin and granzyme B double knock out. Indeed, in human MDSCs, perforin expression was not detected in flow cytometry, while granzyme B was clearly detected. 

As we do not have perforin or granzyme B single knock out mice to our disposal, we used the perforin or granzyme B double knock out mice in subsequent studies, purposed to study whether the expression of these molecules assists in the tumor-promoting function of the MDSCs. We clearly showed a difference in tumor growth in vivo, even if MDSCs (generated from wild type versus perforin/granzyme B double knock outs) were only co-injected in these mice. We further showed in vitro that there was a difference in stimulating tumor growth in the so-called scratch assay, suggesting that influencing the ability of tumor cells to grow (proliferate or infiltrate?) is one way in which wild type MDSCs differ from perforin and granzyme double knock out MDSCs. Literature already describes a role for granzymes (however not perforins) in the growth and metastasis process of tumors (as elaborated on in the introduction of our manuscript), therefore, our observations can be explained by the presence of granzyme B, although an effect of perforin cannot be excluded completely. We have clarified this in the manuscript text.

4.  Figure. 4, authors checked tumor and growth curve at day 5 and around 2 weeks. Due to the in vivo study they use was co-injected of tumor with MDSCs, whether perforin and granzyme B effects directly through MDSCs or indirectly via tumor secreting factors? MDSCs conditional knockout in vivo model and transwell assay should be considered to address direct or contact dependent effect of MDSCs.  

Reply: We agree with the reviewer that an in vivo experiment with a conditional MDSC knock out model would be ideal. To establish a myeloid cell knock out, we studied the CD11b-DTR model, as this one could be obtained through our network of collaborators. Herein myeloid cells can be depleted by injection of diphtheria toxin (DT). However, when optimizing the depletion protocol (amount of DT), we noticed that the dose needed for depletion, had a detrimental effect on the mice’s overall well-being with weight loss >20% in a matter of days. As this is not allowed by our ethical committee, we could not proceed with this model. 

We also considered adding a transwell when performing the scratch assay. However, this interferes with the hourly images that are taken from the growing tumor cells. As we work with cells that are imbedded in matrigel, it is a challenging task to set the focus of the camera correct. With a transwell, the focus is always on the bottom of the transwell rather that the growing tumor cells. Therefore, we have not continued with this idea. Finally, I wish to say that in the time frame provided to perform the revision, it was not feasible to (1) get ethical clearance (takes 2 months at our institute) to kill mice for bone marrow collection and subsequent MDSC differentiation, (2) perform the scratch assay and for instance take a picture before addition of the transwell and at the end point (when removing the transwell). If the reviewer insists on performing this experiment, we would like to ask the editor to extend the time for revision.

Reviewer 2 Report

The manuscript by Dufait et al examines the role of perforin and Granzyme Bon MDSC function. They argue that the expression of these molecules is central to the role of MDSCs in promoting tumor growth and metastasis.

Line 106-110 indicates that MDSCs in patients and healthy donors do not express perforin but do express Granzyme B (Figure 2c). However, the left panel of Figure 2c is inconclusive due to the background as a result of the isotype control. Is there a positive control to show that perforin can be detected by the reagent used? Otherwise, the conclusion that perforin is not expressed by the MDSCs cannot be supported.

Line 132-134: While the authors checked various targets such as Arg-1, iNOS, and MMP9, not all the relevant targets have been checked. For example, IL-10 and TGFb were not checked. The authors themselves note in the discussion that GzmB "has been implicated in cleavage of no less than 10 proteins found within the ECM". As such, it is premature to conclude that "any effects observed in vivo would be due to the effect of perforin and GzmB...". There could still be other factors secreted by MDSCs that result in their ability to facilitate tumor growth. In short, i would suggest changing"would" in line 133 to "could".

Lines 166-174: The conclusions are not supported by the data. While the presence of WT MDSCs improve tumor cell migration, the presence of a granzyme B inhibitor does not seem to effect the impact of WT MDSCs in tumor cell migration. In Figure 5b, it would help to see if the wound width for CTR bar is a significant change compared to the Z-AAD-CMK bar. If it is, then one can argue that factors other than granzyme B is/are influencing tumor cell migration. This data is not provided. If the change in wound width is now not significant, then it is possible that granzyme B is playing a role. 

Author Response

Reviewer 2:

Review Report Form:

Yes

Can be improved

Must be improved

Not applicable

Does the introduction provide sufficient background and include all relevant references?

(x)

( )

( )

( )

Is the research design appropriate?

(x)

( )

( )

( )

Are the methods adequately described?

(x)

( )

( )

( )

Are the results clearly presented?

(x)

( )

( )

( )

Are the conclusions supported by the results?

( )

(x)

( )

( )

Reply: We thank the reviewer for reading and commenting on the manuscript, submitted to “Cancers”. We have done everything possible before the revision deadline of April 29, 2019 to comply with the reviewer’s comments, and provide a more in depth description of these changes in the point-by-point reply to the comments and suggestions of the reviewer.

Comments and Suggestions for Authors:

The manuscript by Dufait et al examines the role of perforin and Granzyme Bon MDSC function. They argue that the expression of these molecules is central to the role of MDSCs in promoting tumor growth and metastasis.

Line 106-110 indicates that MDSCs in patients and healthy donors do not express perforin but do express Granzyme B (Figure 2c). however, the left panel of Figure 2c is inconclusive due to the background as a result of the isotype control. Is there a positive control to show that perforin can be detected by the reagent used? Otherwise, the conclusion that perforin is not expressed by the MDSCs cannot be supported.

Reply: We did not include a positive control, for example stimulated CD8+T-cells, in this experiment. If the reviewer insists on performing this experiment, we would like to ask the editor to extend the time for revision. Meanwhile, we have changed line 122-123 in the results section and line 261-262 and 271 in the discussion. 

Line 132-134: While the authors have checked various targets such as Arg-1, iNOS and MMP9, not all the relevant targets have been checked. For example, IL-10 and TGFb were not checked. The authors themselves note in the discussion that GzmB “has been implicated in cleavage of no less than 10 proteins found in the ECM”. As such, it is premature to conclude that “any effects observed in vivo would be due to the effect of perforin and GzmB…”. There could still be other factors secreted by MDSCs that result in their ability to facilitate tumor growth. In short, I would suggest changing”would” in line 133 to “could”.

Reply: We agree with the reviewer and as requested have changed “would” to “could” in line 147

Lines 166-174: The conclusions are not supported by the data. While the presence of WT MDSCs improve tumor cell migration, the presence of a granzyme B inhibitor does not seem to effect the impact of WT MDSCs in tumor cell migration. In Figure 5b, it would help to see if the wound width for CTR bar is a significant change compared to the Z-AAD-CMK bar. If it is, then one can argue that factors other than granzyme B is/are influencing tumor cell migration. This data is not provided. If the changes in wound width is now not significant, then it is possible that granzyme B is playing a role.

Reply: We apologize for not including this data in the figure. Only one condition is statistically significant in Figure 5, namely Control versus WT MDSCs. CTR versus Z-AAD-CMK results in a p-value of 0,0817. We have modified the indication of the statistical significances in Figure 5b and modified the text to make this observation more clear (line 186-187). We did not alter the conclusions for this section, as the interpretation from this experiment  remain unaltered.

Reviewer 3 Report

This is an intriguing work depicting a relatively new and unexpected functional pathway mediating the immunosuppressive activity of MDSC in cancer. Because of the novelty of the topic here described, some issues must be addressed  more in depth and additional information should be provided.

 Major points

The expression of Granzyme B and perforin should be depicted more extensively in MDSC subsets by showing original facs  dotpots and gating strategies both in mice and patients, possibly including not only MDSC but also other myeloid cell subsets for comparison. The expression in activated CD8+ T cells should be shown for comparison. It would be key seeing the expression in myeloid cells from tumor bearing vs  tumor –free mice, to be confirmed also by PCR in sorted populations.

it is sincerely a bit unexpected that MDSC from perforin/granzyme KO mice lose completely their protumor activity with respect to WT MDSC. Unexpected is also that WT MDSC exert no protumor effect  when coinjected in vivo (figure 4b). Although the authors point to the role of granzymeB/perforin in matrix remodeling, I would like to remind that MDSC  protumor activity relies also on their  ability to induce neoangiogenesis and EMT promotion, along with metastatic spreading. Maybe, the protumor activity of KO vs WT MDSC  should be tested  also under other co-injection schedules or conditions (different cell # and ratios, different time points…), to allow the other MDSC functional pathways to emerge. This would provide a more realistic scenario or, eventually, it would prove the key role of perforin/granzyme in the process.

In line with the abovementioned queries, it would be crucial to demonstrate  that MDSC from KO mice are similar in  frequency and  in vivo distribution with respect to MDSC from WT animals

According to the data in figure 6 B and C, no major difference is instead detectable when the immunosuppressive activity of MDSC from KO vs WT mice on T cells is tested in vitro. At this regard, a titration of the MDSC:T cell ratios should be included to explore whether differences might emerge. Instead, differences are detected in vivo in terms of tumor T cell infiltrate level and activity. As this implies that the granzymeB/perforin pathway is involved mostly in the direct protumor effect of MDSC, it would be interesting showing that these differences are maintained also in the absence of T cells such as in immunodeficient mice.

Minor points

Please include in the manuscript the data obtained in other tumor models that are now mentioned as not shown (page 2 line 90)

It would be nice to see some immunohistochemistry images depicting the increase T cell infiltrate in tumor lesions from figure 6 (e-f)

Please explain better the meaning of the experiments depicted in figure 4A

please increase the size and the graphic quality of some figures, which are hard to read in the present form

Author Response

Reviewer 3:

Review Report Form 

Yes

Can be improved

Must be improved

Not applicable

Does the introduction provide sufficient background and include all relevant references?

( )

(x)

( )

( )

Is the research design appropriate?

( )

( )

(x)

( )

Are the methods adequately described?

( )

( )

(x)

( )

Are the results clearly presented?

( )

(x)

( )

( )

Are the conclusions supported by the results?

( )

( )

(x)

( )

Reply: We thank the reviewer for reading and commenting on the manuscript, submitted to “Cancers”. We have done everything possible before the revision deadline of April 29, 2019 to comply with the reviewer’s comments, and provide a more in depth description of these changes in the point-by-point reply to the comments and suggestions of the reviewer.

Comments and Suggestions for Authors:

This is an intriguing work depicting a relatively new and unexpected functional pathway mediating the immunosuppressive activity of MDSC in cancer. Because of the novelty of the topic here described, some issues must be addressed more in depth and additional information should be provided.

Reply: We thank the reviewer for these appreciative words and recognizing the novelty of this work.

Major points:

1.    The expression of Granzyme B and perforin should be depicted more extensively in MDSC subsets by showing original facs dotpots and gating strategies both in mice and patients, possibly including not only MDSC but also other myeloid cell subsets for comparison. The expression in activated CD8+ T cells should be shown for comparison. It would be key seeing the expression in myeloid cells from tumor bearing vs tumor –free mice, to be confirmed also by PCR in sorted populations.

Reply: As requested by both reviewers, we provided a figure showing the gating strategy on MDSCs (see Figure below). This figure was included in the manuscript as well. We unfortunately were not able to generate for instance dendritic cells or macrophages, because of the time frame provided to perform the revision (deadline April 29, 2019). To perform this experiment, we need (1) ethical clearance (takes 2 months at our institute) to kill mice for bone marrow collection and subsequent dendritic cell or macrophage differentiation, (2) perform flow cytometry analysis. If the reviewer insists on performing this experiment, we would like to ask the editor to extend the time for revision. 

2.    it is sincerely a bit unexpected that MDSC from perforin/granzyme KO mice lose completely their protumor activity with respect to WT MDSC. Unexpected is also that WT MDSC exert no protumor effect when coinjected in vivo (figure 4b). Although the authors point to the role of granzymeB/perforin in matrix remodeling, I would like to remind that MDSC protumor activity relies also on their ability to induce neoangiogenesis and EMT promotion, along with metastatic spreading. Maybe, the protumor activity of KO vs WT MDSC should be tested  also under other co-injection schedules or conditions (different cell # and ratios, different time points…), to allow the other MDSC functional pathways to emerge. This would provide a more realistic scenario or, eventually, it would prove the key role of perforin/granzyme in the process.

Reply: We agree with reviewer that the results of the in vivo experiment in which MDSCs were co-injected with MDSCs were unexpected. We expected (at least with MDSCs generated from bone marrow of wild type mice) to observe an earlier onset and effect on tumor progression. In the conditions used this was not the case for the majority of the mice. 

Injecting MDSCs at multiple time points could be a strategy to see the effect of so-called wild type MDSCs. However, the question we wanted to address is whether or not there was a difference between wild type, and granzyme B and perforin double knock out MDSCs in terms of aiding tumor growth. This was clearly the case. Moreover, as we wanted to evaluate whether this was a direct effect of the MDSCs (not as a result of an interplay with other immune cells attracted to the developing tumor), we performed the in vitro assay in which only MDSCs are added to the growing tumors cells. This assay in our opinion shows a role for granzyme B and perforin on the ability of tumor cells to grow. Of course, this assay does not allow us to claim that it is a key role. We still believe that the different modes of action employed by MDSCs act in concert to provide the MDSC with maximal tumor-promoting capacities.

3.    In line with the abovementioned queries, it would be crucial to demonstrate that MDSCs from KO mice are similar in frequency and in vivo distribution with respect to MDSC from WT animals.

Reply: During the in vivo experiment, we refrained from isolating tumors and performing flow cytometry-based or immunohistochemistry-based analysis to study MDSC frequency or distribution. The rationale was that the time at which we have resectable tumors of about 100 mm3 (5 to 6 mm diameter) that can be used for further analysis, is around day 14 in groups injected with MDSCs. This is 14 days after the co-injection of MDSCs, and we believe based on viability over time in vitro (for which we refer to prior work) that at that stage the MDSCs are likely no longer present in the tumor, and if cells are still present they are likely representing a small fraction of the total MDSC number, as also endogenous MDSCs will be attracted to the tumor. 

Nonetheless, similar questions arose during our discussions on the work, and we tried to set up a model to deplete myeloid cells in vivo, as this would facilitate the analysis of transferred cells, and could be used in function of other research questions (as explained to reviewer 1). To establish a myeloid cell knock out, we studied the CD11b-DTR model, as this model could be obtained through our network of collaborators. Herein myeloid cells can be depleted by injection of diphtheria toxin (DT). However, when optimizing the depletion protocol (amount of DT), we noticed that the dose needed for depletion, had a detrimental effect on the mice’s overall well-being with weight loss >20% in a matter of days. As this is not allowed by our ethical committee, we could not proceed with this model and abandoned the idea of evaluating the transferred cells in an in vivo model.

We further discussed using 3D tumor models to study, growth rate, MDSC location in these 3D spheres. To date we have invested in setting up 3D model and evaluating the variability in growth rate, depending on the ratio tumor cells and fibroblasts, as well as on how we can best analyze tumor growth (confocal microscopy evaluating the area of green fluorescence using green fluorescent tumor cells, digesting the tumor to single cell suspension followed by cell count either manually or using flow cytometry). These experiments have shown high variability, making us doubt whether this system allows evaluating tumor growth after treatment, in this case developing 3D tumors in the presence of MDSCs. We could use this system to evaluate whether the MDSCs are present at a different location. However, this would be highly experimental and probably received with a lot of skepticism if differences are observed. We make this claim as we noticed with every study using the in vitro generated MDSCs that a least one reviewer is skeptical. We understand this skepticism, but fear that using two in vitro models that are not generally accepted will not aid in convincing scientists of potential differences in MDSC location. Nonetheless, if the reviewer insists that we perform this experiment, we will (1) request ethical clearance (takes 2 months at our institute) to kill mice for bone marrow collection and subsequent MDSC (in and macrophage (IFN-gamma or IL-4 protocol) differentiation, (2) set up 3D tumors for analysis of MDSC versus macrophages, (3) perform microscopy analysis of their location. We would take along macrophages exposed to IFN-gamma versus IL-4 as these show a different location in tumors (non-hypoxic versus hypoxic regions), which will allow us to confirm or not that the 3D cultures can be used to study location of myeloid cells. If the reviewer insists on performing this experiment, we would like to ask the editor to extend the time for revision. 

4.    According to the data in figure 6 B and C, no major difference is instead detectable when the immunosuppressive activity of MDSC from KO vs WT mice on T cells is tested in vitro. At this regard, a titration of the MDSC:T cell ratios should be included to explore whether differences might emerge. Instead, differences are detected in vivo in terms of tumor T cell infiltrate level and activity. As this implies that the granzymeB/perforin pathway is involved mostly in the direct protumor effect of MDSC, it would be interesting showing that these differences are maintained also in the absence of T cells such as in immunodeficient mice.

Reply: In the in vitro MDSC suppression assay, we used a 1 on 2 ratio, because at this ratio the suppression by in vitro generated MDSCs is high enough to show a reduction in suppressive activity (which we expected). While discussing this comment, we came to the decision of not repeating this experiment, as we believe that the suppressive activity of MDSCs using this experimental set up will not be changed because also the activated T cells are likely to produce perforin and granzyme B, which likely masks the impact of the absence of perforin and granzyme B in granzyme B and perforin double knock out MDSCs. Therefore in hindsight, this assay could be considered flawed to address the question whether MDSCs generated from wild type versus granzyme B and perforin double knock mice have direct effects on T cells. We will address this in the revised manuscript.

Minor points:

1.    Please include in the manuscript the data obtained in other tumor models that are now mentioned as not shown (page 2 line 90)

Reply: We included the following data:

- Perforin and GzmB expression in both M- and PMN-MDSCs (Figure 1b).

2.    It would be nice to see some immunohistochemistry images depicting the increase T cell infiltrate in tumor lesions from figure 6 (e-f)

Reply: We unfortunately did not prepare material for immunohistochemical analysis, and as mentioned earlier we could not perform this work in the given time frame of revision due to the time needed to get permission to perform this experiment from our ethical committee and time needed to actually perform the experiment.

3.    Please explain better the meaning of the experiments depicted in figure 4A

Reply: As requested we have clarified the experiment and its results depicted in figure 4A.

4.    Please increase the size and the graphic quality of some figures, which are hard to read in the present form

Reply: We wonder whether the graph quality was changed due to conversion (mac – pc, word – pdf) as in our version everything is clear. We will send the original files (tiffs) to the editor.

Round 2

Reviewer 1 Report

To fit the title ” Perforin and Granzyme B Expressed by Murine Myeloid-Derived Suppressor Cells Play a Role in Outgrowth of Cancer Cells”, authors should consider to provide data of perforin that source from MDSC to enhance tumor growth directly via transwell invasion and in vivo assay with image pictures. Otherwise, only Fig. 4b could provide partial information of perforin expressed by MDSC play roles in tumor growth indirectly which cannot support title.

Author Response

To fit the title ” Perforin and Granzyme B Expressed by Murine Myeloid-Derived Suppressor Cells Play a Role in Outgrowth of Cancer Cells”, authors should consider to provide data of perforin that source from MDSC to enhance tumor growth directly via transwell invasion and in vivo assay with image pictures. Otherwise, only Fig. 4b could provide partial information of perforin expressed by MDSC play roles in tumor growth indirectly which cannot support title.

We appreciate the reviewer's comment regarding the role of perforin in our study. We provide information on expression of perforin and granzyme B by mouse MDSCs (Figure 1). To study the significance of these molecules in the mouse setting, we use MDSCs generated starting from bone marrow of C57BL/6 versus C57BL6 perforin and granzyme B double knock out mice, as we unfortunately could not obtain single knock out models. We showed that the perforin and granzyme B double knock MDSCs differed from wild type MDSCs in several assays, showing that in the presence of wild type MDSCs tumors grow faster  (Figure 4), tumor cells have an enhanced capacity to migrate in the scratch assay (Figure 5) and that tumors contain less CD8+ T cells (Figure 6). In these assays, we can not define whether the effects could be attributed to perforin, granzyme B or both. However, in the in vitro scratch assay we did take along a granzyme B inhibitor as a means to identify the need for granzyme B versus perforin, showing that inhibiting only granzyme B partially restored the migratory capacity, and as such indeed arguing against a role of perforin in enabling migration of tumor cells. Nonetheless, in the in vivo studies, in particular the study showing lower CD8+ T cell infiltration in tumors, we can not exclude a role for perforin on the observed effects. 

In order to comply with the reviewer's comments, we have revised the manuscript and placed emphasis on granzyme B when required. These changes have been highlighted in yellow for the reviewer's convenience. We moreover, changed the title of the manuscript to "Perforin and Granzyme B Expressed by Murine Myeloid-Derived Suppressor Cells: a Study on their  Role in Outgrowth of Cancer Cells”, as such we indicate that we studied their role without claiming that both play an equal role in outgrowth of cancer cells.

Reviewer 2 Report

No comments

Author Response

No comments

We appreciate the reviewers effort to revise our submitted manuscript and are pleased that we were able to address all raised comments

Reviewer 3 Report

I appreciate the authors efforts to address all the raised queries and concerns

Author Response

I appreciate the authors efforts to address all the raised queries and concerns

We appreciate the reviewers effort to revise our submitted manuscript and are pleased that we were able to address all raised comments

Round 3

Reviewer 1 Report

Please kindly provide tumor growth image data (or IHC stained with MDSC pattern) to support Figure 4b.